Brief Communication

# Anoxia activates CRISPR–Cas immunity in the mouse intestine

Ian W. Campbell [1,2,5] ✉, David W. Basta [3,5], Franz G. Zingl[1,2], Emily J. Sullivan [1,2], Sudhir Doranga [1,2] & Matthew K. Waldor [1,2,4] ✉

The natural context in which CRISPR–Cas systems are active in Enterobacteriaceae has remained enigmatic. Here we find that the *Citrobacter rodentium* type I-E CRISPR–Cas system is activated by the oxygen-responsive transcriptional regulator Fnr in the anoxic environment of the mouse intestine. Since Fnr-dependent regulation is predicted in ~41% of Enterobacteriaceae *cas3* orthologues, we propose that anoxic regulation of CRISPR–Cas immunity is an adaptation that protects Enterobacteriaceae against threats from foreign DNA within the intestinal microbiome.

Prokaryotes use clustered regularly interspaced short palindromic repeats and CRISPR-associated protein (CRISPR–Cas) systems to recognize and cleave foreign nucleic acid sequences to protect against phages and other mobile genetic elements[1–3]. However, with a few exceptions[4,5], little is known about the regulation of these systems under physiological conditions. The limited understanding of CRISPR–Cas regulation is partly attributable to the absence of native CRISPR–Cas activity in cultured Enterobacteriaceae, a commonly studied family of bacteria, necessitating investigation using artificial overexpression systems[6–8].

*Citrobacter rodentium* is a Gram-negative bacterial pathogen that naturally infects and causes colitis in mice[9]. Like most Enterobacteriaceae, *C. rodentium* is a facultative anaerobe, capable of growth in the presence or absence of oxygen (oxic or anoxic conditions, respectively). We performed RNA sequencing of the pathogen's transcriptional response to anoxia and observed more transcripts from the type I-E *cas* locus (schematized in Fig. 1a) in anoxic versus oxic culture conditions (Fig. 1b, Extended Data Fig. 1 and Supplementary Table 1). To test whether increased *cas* expression correlates with CRISPR–Cas activity, we designed a functional assay[10] that monitors the retention frequency of either a plasmid containing a sequence that is (target) or is not (control) recognized by the native *C. rodentium* CRISPR locus (plasmid retention assay; schematized in Extended Data Fig. 2).

In oxic culture conditions, 92% of cells retained the target plasmid, indicating that CRISPR–Cas immunity is inactive (Fig. 1c). By contrast, only 1% of cells retained the target plasmid during anoxic culture (Fig. 1c), demonstrating anoxic-specific immunity. To verify that this assay requires CRISPR–Cas activity, we deleted the entire *C. rodentium cas* locus (Δ*cas3-2*) and repeated the assay. Deletion of

*cas3-2* eliminated CRISPR–Cas immunity during anoxic culture (Fig. 1d). We conclude that anoxia is required for both expression and activity of *C. rodentium* CRISPR–Cas immunity.

To discover anoxic regulators of the *C. rodentium cas* locus, we leveraged the plasmid retention assay for a transposon-insertion loss-of-function screen (InducTn-seq[11]). We anoxically cultured a transposon mutant population of *C. rodentium* carrying the target plasmid and then sequenced the mutants that retained the plasmid following antibiotic selection, comparing with a population containing a control plasmid (Fig. 1e and Supplementary Table 2). As expected, transposon insertions in the endonuclease *cas3* were specifically enriched in the population retaining the target plasmid, indicating that disruption of *cas3* prevents CRISPR–Cas immunity. By contrast, *cas1* and *cas2*, which are not involved in CRISPR–Cas interference, were not enriched, demonstrating the specificity of the assay. One of the two most enriched non-*cas*-related genes was *fnr*, which encodes an oxygen-responsive transcriptional regulator widely conserved among facultative anaerobic bacteria[12] (Fig. 1e). The other most enriched gene was a chaperone involved in the maturation of iron–sulfur cluster-containing proteins (*hscA*), previously demonstrated to be needed for full Fnr activity[13]. Based on these data, we hypothesized that Fnr regulates CRISPR–Cas expression.

In support of this hypothesis, CRISPR–Cas immunity was eliminated in the absence of *fnr* (Δ*fnr*; Fig. 1d). Mutation of a putative Fnr-binding motif[14,15] centred 69.5 nucleotides upstream of *cas3* (Fig. 1f) also eliminated CRISPR–Cas immunity (Fig. 1d). Furthermore, quantitative PCR (qPCR) demonstrated that mutation of the Fnr-binding site upstream of *cas3* eliminated the transcriptional response of *cas3*

[1]Division of Infectious Diseases, Brigham and Women's Hospital, Boston, MA, USA. [2]Department of Microbiology, Harvard Medical School, Boston, MA, USA. [3]Department of Pathology, Brigham and Women's Hospital, Harvard Medical School, Boston, MA, USA. [4]Howard Hughes Medical Institute, Boston, MA, USA. [5]These authors contributed equally: Ian W. Campbell, David W. Basta. ✉e-mail: icampbell3@bwh.harvard.edu; mwaldor@bwh.harvard.edu

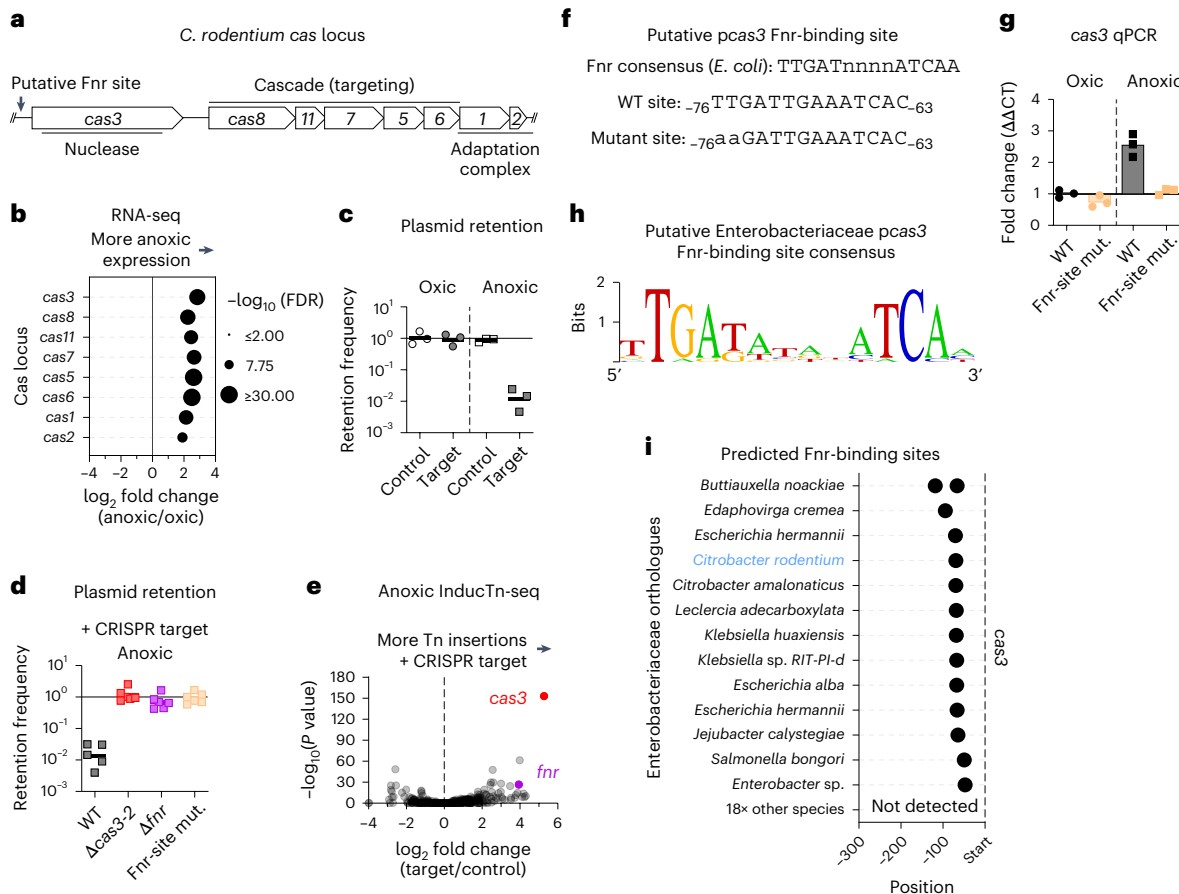

**Fig. 1 | Anoxia causes Fnr-dependent activation of CRISPR–Cas immunity in *C. rodentium*. a**, A schematic of the *C. rodentium cas* locus. **b**, RNA sequencing (RNA-seq) results from the *cas* locus of *C. rodentium* cultured under oxic or anoxic conditions for 3.5 h on solid LB agar. Data represent three biological replicates. FDR, false discovery rate. Additional analysis in Extended Data Fig. 1 and Supplementary Table 1. **c,d**, Fraction of cells from a single colony cultured for 24 h on solid LB agar that retained the CRISPR–Cas target or control plasmid. Assay design in Extended Data Fig. 2. Geometric mean of biological replicates. For **c**: 3 colonies each. For **d**: wild-type (WT), 5 colonies; Δ*cas3-2*, Δ*fnr*, and Fnr-site mut. (with mutation displayed at the bottom of **f**), 6 colonies. **e**, InducTn-seq volcano plot comparing fold change in the gene insertion frequency between cells that retained the CRISPR target or control plasmid. Points represent individual genes. *P* value from two-sided Mann–Whitney *U* test. Additional data in Supplementary Table 1. **f**, Fnr-binding sites. Top: consensus defined in *E. coli*[15]. Bottom: putative *C. rodentium* WT and mutated Fnr-binding sites relative to the *cas3* start codon. **g**, qPCR of *cas3* following 3.5 h of culture on solid LB agar. ΔΔCT (threshold cycle) analysis compared with *rpoA* and oxic culture conditions. Data represent three biological replicates (plates) with three technical replicates per sample. **h,i**, Consensus motif[28] (**h**) and location relative to *cas3* start codon (**i**) of putative Fnr-binding sites upstream of Enterobacteriaceae *cas3* orthologues.

to anoxia (Fig. 1g). These results indicate that Fnr directly activates CRISPR immunity during anoxia.

To determine the conservation of Fnr-mediated regulation of CRISPR–Cas immunity, we used OrthoDB[16] to select 501 non-redundant Gammaproteobacteria genomes containing *cas3* orthologues and interrogated the 300 nucleotides upstream of *cas3* with motif enrichment analysis[14]. In total, 141 of 501 genomes contained at least one predicted Fnr-binding site upstream of *cas3* (Supplementary Tables 3 and 4). These genomes were distributed among most orders of Gammaproteobacteria (Extended Data Fig. 3). Notably, the Fnr-binding sites in Enterobacteriaceae closely matched the Fnr-consensus sequence previously defined in *Escherichia coli* strain MG1655 (ref. 15) and were primarily centred at the same position as in *C. rodentium* (Fig. 1h,i and Extended Data Fig. 4). The positional conservation of an Fnr-binding motif in 13 out of 32 Enterobacteriaceae suggests conserved Fnr-dependent *cas3* activation within a subset of this family.

Many of the Enterobacteriaceae with an Fnr-binding motif upstream of *cas3* have been isolated from the mammalian intestine (for example, *Escherichia*, *Citrobacter* and *Klebsiella*), which is frequently an anoxic environment[17]. We hypothesized that activation of CRISPR–Cas immunity by anoxia may protect Enterobacteriaceae from threats encountered within the microbially rich intestine. Consistent

with this hypothesis, RNA sequencing revealed that faecal-associated *C. rodentium* isolated from infected C57BL/6J mice had more transcripts from the *cas* locus than in oxic culture (Fig. 2a, Extended Data Fig. 5 and Supplementary Table 1).

To determine if increased *cas* expression results in CRISPR–Cas immunity within the intestine, we infected mice with *C. rodentium* strains carrying the CRISPR–Cas target plasmid and monitored plasmid retention in faecal bacteria. Over the first 24 h, most shed bacteria retained the plasmid (Fig. 2b). Subsequently, wild-type *C. rodentium* progressively lost the plasmid, with only 1% of shed bacteria retaining the plasmid 13 days after inoculation. By contrast, 85% of Δ*cas3-2* cells and 23% of Δ*fnr* cells retained the plasmid over the same 13-day period (Fig. 2b). These results suggest that anoxia within the intestine results in Fnr-dependent activation of CRISPR–Cas immunity in *C. rodentium*. Furthermore, the discrepancy between the Δ*cas3-2* and Δ*fnr* mutants suggests that intestinal signals beyond anoxia may regulate CRISPR activity—potentially by relieving H-NS-mediated transcriptional repression, as previously observed in other bacterial species[7,8,18].

The intestine contains low concentrations of oxygen and dense communities of microbes and phage[17,19]. Density-dependent regulation is consistent with previous reports of quorum-sensing-regulated CRISPR–Cas activity in *Serratia* spp., *Pseudomonas aeruginosa* and

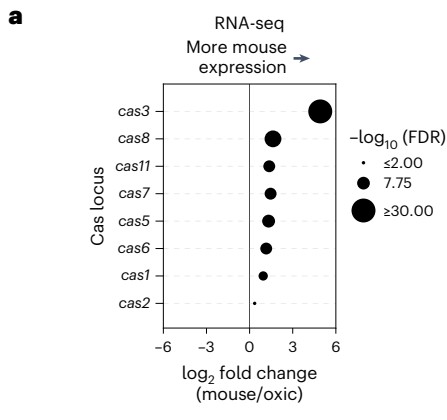

**a**

RNA-seq
More mouse
expression →

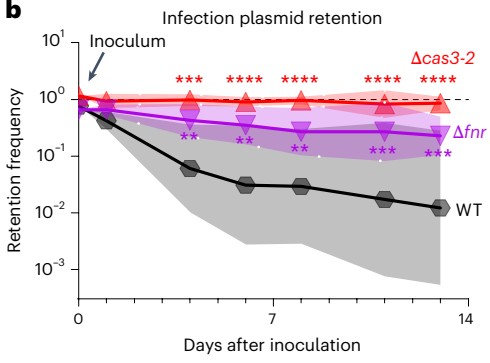

**b**

**Fig. 2 | _C. rodentium_ CRISPR–Cas immunity is activated by Fnr within the murine intestine. a**, Cas transcripts from RNA sequencing (RNA-seq) of _C. rodentium_ recovered from the faeces of infected, female, C57BL/6J mice 7 days after inoculation compared with bacteria from oxic culture. Data represent three biological replicates from mice and three from oxic culture. Additional analysis in Extended Data Fig. 5 and Supplementary Table 1. **b**, Retention of a CRISPR target plasmid by the indicated strains of _C. rodentium_ measured by serial dilution and plating of faeces from infected mice ($N = 16$ wild type (WT), 8 $\Delta fnr$, 8 $\Delta cas3$-$2$ infected mice; equal mix of sexes). Lines represent the geometric mean, and shading represents the geometric standard deviation. Significance compared with wild-type by two-way analysis of variance with Dunnett's multiple-comparison test on $\log_{10}$-transformed data (**$P \leq 0.01$, ***$P \leq 0.001$, ****$P \leq 0.0001$).

_Aliivibrio wodanis_[19–21]. We therefore propose that, in dense microbial communities such as the host intestine, anoxic regulation of CRISPR–Cas immunity in _C. rodentium_ and other Enterobacteriaceae represents an adaptation that protects these bacteria against predation.

## Methods

### Regulatory statement

All bacterial work was performed in biosafety level 2 facilities at the Brigham and Women's Hospital according to protocols reviewed and approved by the Brigham and Women's Hospital Institutional Biosafety Committee under protocol 2011B000082. All personnel working with bacteria were trained in relevant safety and protocol-specific procedures. Animal studies were conducted at Brigham and Women's Hospital in compliance with the 'Guide for the Care and Use of Laboratory Animals' and according to protocols reviewed and approved by the Brigham and Women's Hospital's Institutional Animal Care and Use Committee under protocol 2016N000416.

### Bacterial strains

The strains used in this study are listed in Supplementary Table 5. _C. rodentium_ is a spontaneous streptomycin-resistant isolate of strain ICC168 (ref. [22]), previously known as _Citrobacter freundii_ biotype 4280 (ATCC 51459). _Escherichia coli_ strain MFD_pir_ was used for cloning[23].

### Oxic and anoxic culture

Bacteria were cultured at 37 °C in either liquid lysogeny broth (LB) shaking at 200 rotations per minute or on solid LB containing 1.5% agar. Oxic culture was performed in atmospheric conditions. Anoxic culture was performed in a Baker Concept 400M anaerobic workstation set to 0% oxygen, with media acclimated to the anoxic environment for at least one night.

### Plasmid assembly

Plasmid fragments were amplified from plasmid or genomic DNA using single-stranded DNA primers (Integrated DNA Technologies). Fragments were assembled with NEBuilder HiFi DNA Assembly Master Mix (New England Biolabs). Assembled plasmids were transformed into _E. coli_ strain MFD_pir_ with electroporation and transferred to the recipient strains by conjugation.

### Constructing mutant _C. rodentium_ strains

The allelic exchange protocol from ref. [24] was used to create in-frame deletions, and the Fnr-site mutation in _C. rodentium_. pTOX5 (Genbank MK972845) was linearized with the restriction enzyme SwaI and assembled with ~1-kb homology arms flanking the desired mutation. Primer sequences are included in Supplementary Table 6. For deletions, two to three codons were left intact at both ends of the deletion. This plasmid was electroporated into MFD_pir_, checked by PCR and conjugated into _C. rodentium_. Transconjugants were purified by plating consecutively three times. Individual colonies were cultured for 1 h without antibiotic selection and then counterselected to isolate double crossovers lacking the plasmid backbone. Single colonies were selected, and PCR and whole-genome sequencing confirmed the identity and fidelity of the mutant strains.

### Animal experiments

Adult (9–12 weeks) C57BL/6J mice were purchased from Jackson Laboratory (strain #000664) and acclimated for at least 72 h before experimentation. During infection, mice were housed under specific pathogen-free conditions at 68–75 °F, with 30–50% humidity and a 12-h light/dark cycle in a biosafety level 2 facility.

For _C. rodentium_ infections, mice were deprived of food for 3–5 h before inoculation. Animals were then mildly sedated with isoflurane, and 100 μl of the indicated strain suspended in phosphate-buffered saline (PBS) was inoculated into the stomach with a sterile feeding needle (Cadence Science). Dose (streptomycin-resistant colony-forming units, CFU) and CRISPR plasmid retention (gentamicin-resistant CFU) were determined retrospectively by serial dilution and plating.

Animal health was monitored during infection by measuring weight, body condition and faecal appearance. _C. rodentium_ colonization and plasmid retention were monitored by sampling faeces from infected animals. Fresh faecal pellets were suspended in sterile PBS and homogenized in a bead beater (BioSpec Products) with 3.2-mm stainless-steel beads. _C. rodentium_ concentration (CFU g$^{-1}$) and plasmid retention were determined by serial dilution and plating for CFUs.

### RNA sequencing

For cultured samples, _C. rodentium_ was grown overnight in liquid LB in oxic conditions. A total of $4 \times 10^7$ CFU from the stationary phase culture were seeded onto LB agar plates and cultured for 3.5 h in the presence or absence of oxygen. Cells were immediately diluted in two parts Qiagen RNAprotect Bacteria Reagent and stored at −80 °C until processing.

For faecal samples, female mice were infected with $5 \times 10^9$ CFU of _C. rodentium_, and colonization was monitored in the faeces to ensure engraftment. Seven days after inoculation, fresh faeces from infected mice were submerged in RNAprotect Bacteria reagent and manually disrupted with a sterile rod to release bacteria. To remove large debris and eukaryotic cells, samples were passed through a 5-μm filter and bacteria were frozen at −80 °C until processing.

RNA was released from bacteria using lysozyme and proteinase K digestion. RNA was extracted with a Qiagen RNeasy kit and purified with RNA Clean & Concentrator-5 (Zymo Research). SeqCenter performed library preparation and sequencing using the following method, provided by SeqCenter: "Samples were DNAse treated with Invitrogen DNAse (RNAse free). Library preparation was performed using Illumina's Stranded Total RNA Prep Ligation with Ribo-Zero Plus kit and 10 bp unique dual indices (UDI). Sequencing was done on a NovaSeq X Plus, producing paired end 150 bp reads. Demultiplexing, quality control, and adapter trimming was performed with bcl convert."

RNA sequencing data was processed using CLC Genomics Workbench (Qiagen). RNA-Seq Analysis 2.8 parameters: reference – ICC168 reference genome; mismatch cost = 2; insertion cost = 3; deletion cost = 3; length fraction = 0.8; similarity fraction = 0.8; global alignment; strand specific = reverse; max hits per read = 10; count paired reads as two = no; ignore broken pairs. Differential gene expression was determined with Differential Expression for RNA-Seq 2.8. Results are included in Supplementary Table 1.

### Plasmid retention assay

The assay is schematized in Extended Data Fig. 2. The CRISPR-target plasmid contains the gentamicin resistance gene *aaC1* and a protospacer adjacent motif (PAM; AAG) followed by a protospacer sequence matching the native CRISPR array (ATCTGTTTATAGCTGGCTATAAAATT-TATAAA). The control plasmid is identical except that the protospacer sequence is replaced by a protospacer recognized by the *E. coli* MG1655 CRISPR–Cas system (GCAACGACGGTGAGATTTCACGCCTGACGCTG), and not the *C. rodentium* native CRISPR–Cas system.

For plasmid retention assays in culture, strains carrying the target or control plasmids were outgrown overnight in an oxic environment on solid LB plates with gentamicin. The next day, strains were restreaked onto LB plates without antibiotics and cultured in the presence or absence of oxygen for 24 h. Single colonies were resuspended in sterile PBS, and serial dilution was used to determine the fraction of the population that retained the plasmid (gentamicin-resistant divided by streptomycin-resistant CFU).

For plasmid retention assays during infection, an equal mix of male and female mice were inoculated with ~5 × 10$^9$ CFU of the indicated strain. Plasmid retention was measured in the inoculum and faeces throughout the infection.

### InducTn-seq

Control or CRISPR-target plasmids were conjugated into a *C. rodentium* InducTn-seq mutant library[11]. A total of 3 × 10$^8$ transconjugants were expanded in oxic conditions on LB containing gentamicin, to select for the plasmid, and arabinose, to induce further miniTn5 transposition. The mutant libraries were stored at −80 °C in PBS with 20% glycerol. Subsequently, 5 × 10$^7$ CFU of the mutant libraries were seeded onto LB agar plates and cultured under anoxic conditions for 24 h. The population was then expanded in oxic conditions on LB plates containing gentamicin to select for mutant cells that retained the plasmid during anoxic culture. Cells were frozen at −80 °C until processing.

Sequencing libraries were prepared with the protocol from ref. 11. Genomic DNA was extracted using a DNeasy Blood and Tissue Kit (Qiagen) and sheared to approximately 400 bp using a M220 ultrasonicator (Covaris). The fragmented DNA was then end-repaired using the Quick Blunting Kit (NEB), polyadenylated with Taq polymerase and dATP, and Illumina P7 adapters were ligated using T4 DNA ligase (NEB). The end of the miniTn5 transposon within the integrated InducTn-seq vector was removed by double restriction enzyme digestion followed by SPRIselect size-selection. Transposon-adjacent sequences were amplified from 800 ng of DNA by PCR using an Illumina i7 index sequence on the reverse primer. Primer dimers were removed by size selection, and samples were sequenced on a NextSeq 1000 (Illumina).

InducTn-seq data were analysed with the protocol from ref. 11 using Python. MiniTn5 transposon-insertion frequency was compared between populations that retained the control or target plasmids. Significance was measured with the non-parametric Mann–Whitney *U* statistical test with Benjamini–Hochberg multiple testing correction. Results are included in Supplementary Table 2.

### qPCR

Bacteria were cultured in liquid LB in oxic conditions. Approximately 10$^7$ CFU from the culture were seeded onto LB agar plates and cultured for 3.5 h in the presence or absence of oxygen. After culture, cells were diluted immediately in two parts Qiagen RNAprotect Bacteria reagent and frozen at −80 °C until processing.

RNA was released from bacteria using lysozyme and proteinase K digestion, and extracted using an RNeasy kit (Qiagen). qPCR was performed using the Luna Universal One-Step RT-qPCR kit on a StepOnePlus Real-Time PCR System (Applied Biosystems). At least three biological and three technical replicates were included per sample, with primers targeting both *rpoA* and *cas3* transcripts. Primer sequences are included in Supplementary Table 6.

qPCR data were analysed by comparative critical threshold (CT) analysis. The average *cas3* CT of three technical replicates was first normalized to the CT of *rpoA* from the same sample. Then, the CT was normalized using the *cas3* CT from oxic culture, producing ΔΔCT.

### Phylogenetic analysis of *cas3* orthologues

In total, 578 *cas3* orthologues from 500 Gammaproteobacteria were selected for analysis by OrthoDB (version 12.0)[16]. We added *E. coli* strain EDL933 to this list. Strains are listed in Supplementary Table 3. To construct a phylogeny, we retrieved the nucleotide sequence of *dnaA* from the National Center for Biotechnology Information (NCBI) for 482 of the Gammaproteobacteria and used MAFFT (strategy: FFT-NS-2; model: DNA200; v7.526)[25] to align the sequences, FastTree (model: Jukes-Cantor with CAT rate heterogeneity; v2.1.11)[26] to construct the phylogeny and iTOL (Interactive Tree of Life; version 7.1)[27] to create a visualization.

For motif enrichment analysis, we retrieved 300 bp upstream of *cas3* from NCBI for 553 of the 579 *cas3* orthologues. These sequences were compared with the prokaryotic transcription factor motif database PRODORIC (release 2021.9) with Simple Enrichment Analysis (SEA; version 5.5.7)[14] with the following parameters: differential enrichment analysis; both strands; Fisher exact test; control sequences from shuffled sequences, preserving 3-mer frequencies; hold-out 10% of sequences. This analysis determined that the Fnr motif defined in *E. coli* strain MG1655 (ID MX000004) was significantly enriched within the dataset ($P = 6.87 × 10^{-5}$). The location relative to *cas3* and the scores of putative Fnr-binding sites are included in Supplementary Table 4. The consensus logo of putative Enterobacteriaceae Fnr-binding sites was created with WebLogo version 2.8.2 (ref. 28).

### Software and statistics

Data analysis was performed using CLC genomics workbench (version 24.0.1), GraphPad Prism (version 10.4.1), Python (version 3.12) and Microsoft Excel. The number of samples and statistical tests are described in the figure legends. Graphics were prepared with GraphPad Prism and Microsoft PowerPoint. Reads were mapped to the *C. rodentium* ICC168 genome FN543502.1.

### Licence information

licence immediately upon publication. This Brief Communication is the result of funding in whole or in part by the National Institutes of Health (NIH). It is subject to the NIH Public Access Policy. Through acceptance of this federal funding, the NIH has been given the right to make this Brief Communication publicly available in PubMed Central upon the Official Date of Publication, as defined by the NIH.

## Reporting summary

Further information on research design is available in the Nature Portfolio Reporting Summary linked to this article.

## Data availability

RNA sequencing and InducTn-seq sequencing reads are deposited in the Sequencing Read Archive (SRA) under accession no. PRJNA1254768. Results from RNA-seq analysis are included in Supplementary Table 1. Results from Tn-seq analysis are included in Supplementary Table 2. To request biological materials or information related to this Brief Communication, please contact the corresponding authors. Source data are provided with this paper.

## Code availability

Custom scripts for Tn-seq analysis were originally developed in ref. 11 and are available via GitHub at https://github.com/dbasta27/InducTn-seq.

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

## Acknowledgements

We thank members of the Waldor lab for helpful discussions and feedback on the manuscript. Funding included Howard Hughes Medical Institute (HHMI) and National Institutes of Health (NIH) R01 AI042347 (M.K.W.), Life Science Research Foundation Zingl-2024HHMI (F.G.Z.), Harvard Digestive Disease Center and NIH P30 DK034854 (I.W.C.) and Brigham and Women's Hospital Department of Pathology Stanley L. Robbins Memorial Research Fund Award and NIH T32 AI007061 (D.W.B.). The funders had no role in the study design, data collection and analysis, decision to publish or preparation of the manuscript.

## Author contributions

Conceptualization: I.W.C., D.W.B. and M.K.W. Methodology: I.W.C., D.W.B. and F.G.Z. Investigation: I.W.C., D.W.B., E.J.S. and S.D. Visualization: I.W.C. and F.G.Z. Writing the original draft: I.W.C., D.W.B. and M.K.W. Reviewing and editing the manuscript: I.W.C., D.W.B., F.G.Z., E.J.S., S.D. and M.K.W.

## Competing interests

The authors declare no competing interests.

## Additional information

**Extended data** is available for this paper at https://doi.org/10.1038/s41564-025-02172-8.

**Correspondence and requests for materials** should be addressed to Ian W. Campbell or Matthew K. Waldor.

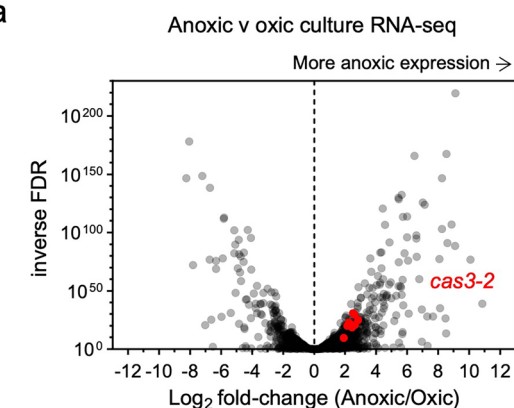

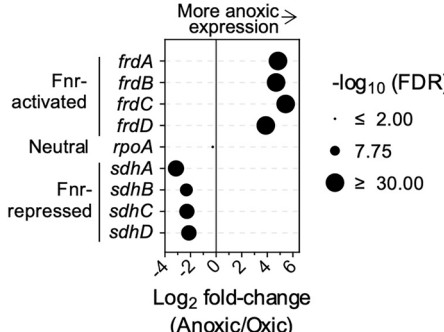

**Extended Data Fig. 1 | Extended presentation of RNA-seq data from Fig. 1.** **a-b**, RNA-sequencing results from *C. rodentium* cultured with (oxic) or without oxygen (anoxic) for 3.5 hours on solid LB agar. 3 biological replicates. False discovery rate, FDR. **b**, The *frd* and *sdh* loci were previously determined to be directly activated or repressed, respectively, by Fnr in anoxic conditions in *Escherichia coli* strain MG1655[15]. *rpoA* is used as a neutral control for qPCR in Fig. 1g. Additional data in Supplementary Table 1.

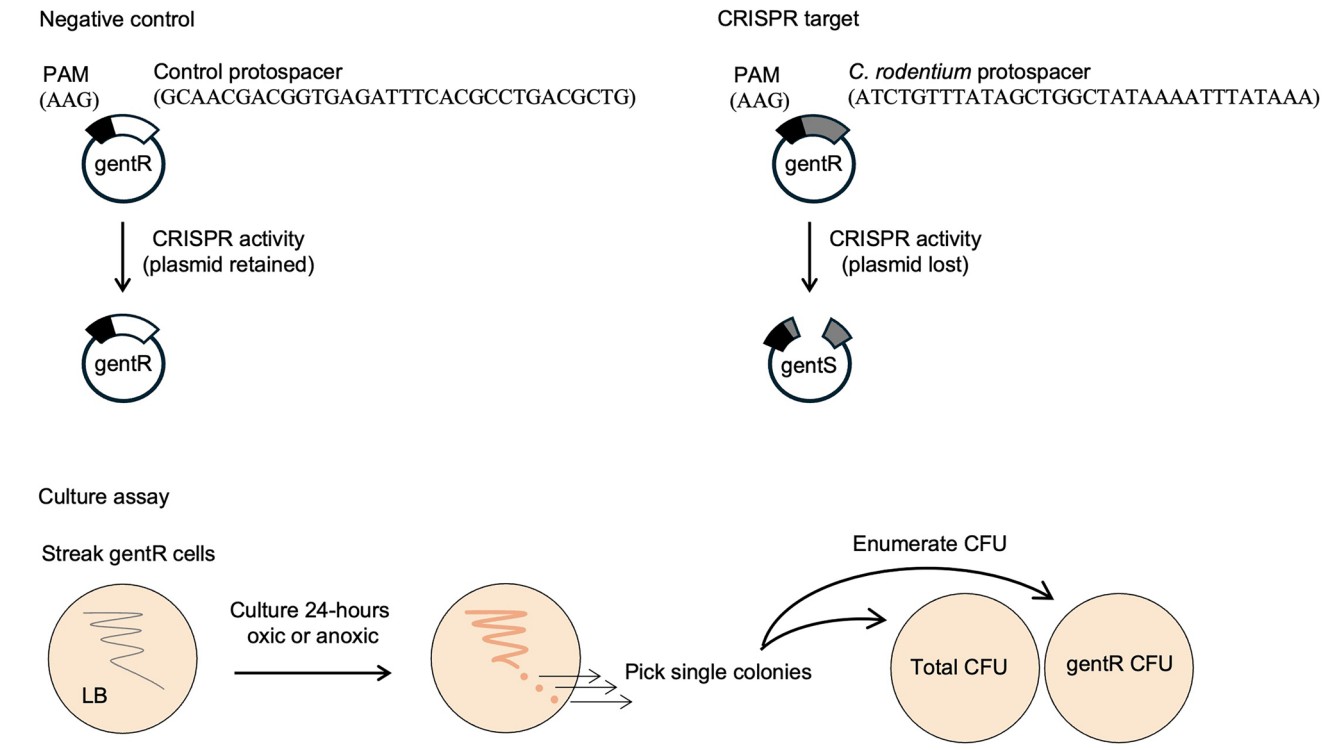

**Extended Data Fig. 2 | Plasmid retention assay.** CRISPR-Cas activity was measured by retention frequency of a CRISPR target plasmid. The CRISPR target plasmid encodes a gentamycin resistance (gentR) gene and a protospacer adjacent motif (PAM) followed by a protospacer sequence matching the native CRISPR array. The control plasmid is identical, except that the protospacer sequence is replaced with a protospacer not recognized by the native CRISPR-Cas system. CRISPR-Cas activity converts cells from gentamycin resistant to gentamycin sensitive (gentS) by causing cleavage and loss of the CRISPR target plasmid, but not the control plasmid. To measure CRISPR-Cas activity in culture, gentamycin resistant cells were streaked onto plates and cultured in oxic or anoxic conditions for 24-hours. After culture, single colonies were picked and total versus gentR CFU was determined by serial dilution.

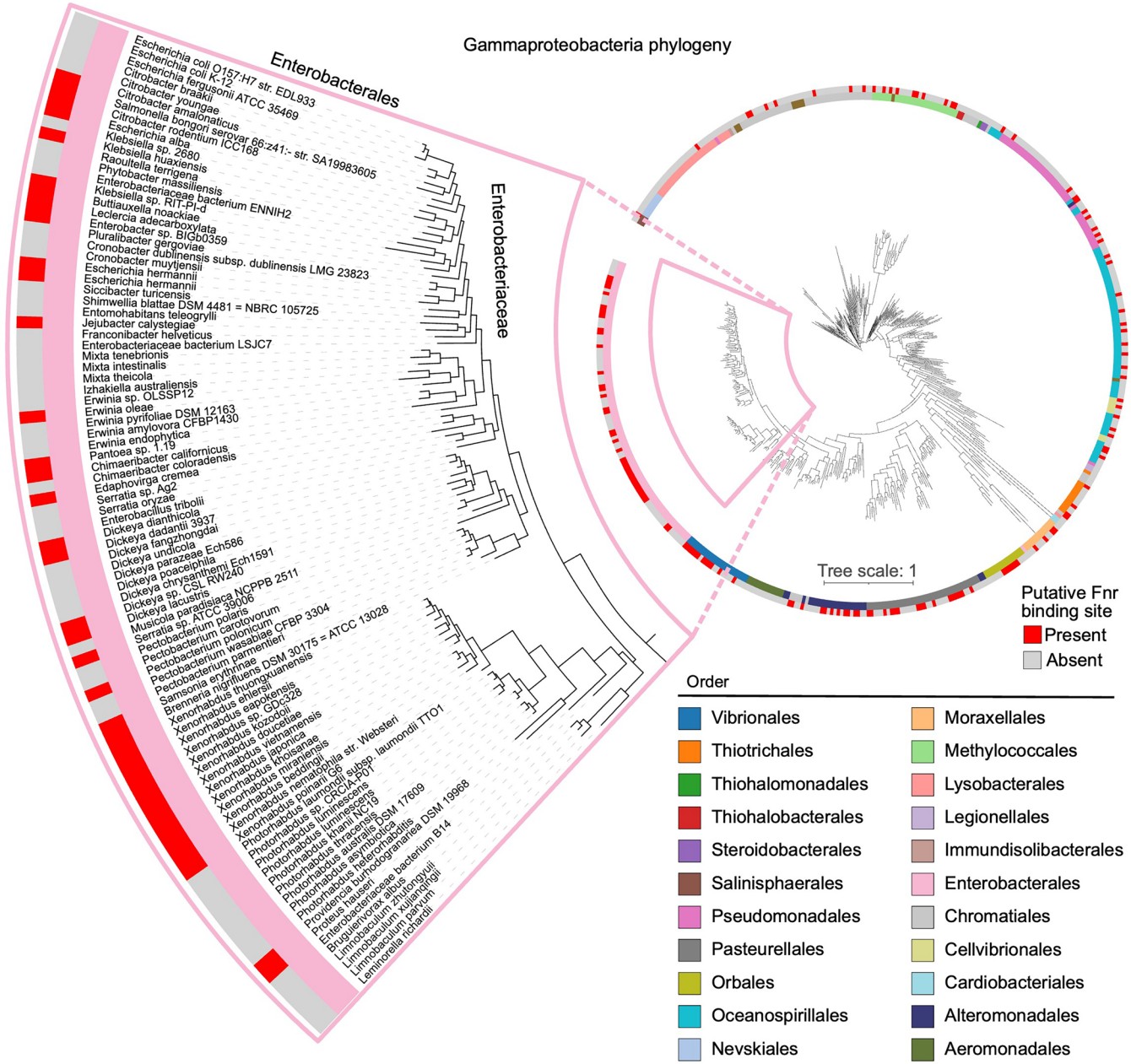

**Extended Data Fig. 3 | Potential Fnr binding sites are present upstream of *cas3* in many Gammaproteobacteria genomes.** Gammaproteobacteria *cas3* orthologs were curated from OrthoDB[16] and the phylogeny was constructed from 482 *dnaA* sequences. Putative Fnr binding sites in the 300 bp upstream of *cas3* were identified with motif enrichment analysis[14] with the Fnr motif defined in *E. coli* strain MG1655. Tree scale is the number of substitutions per site. Strains are listed in Supplementary Table 3 and putative Fnr binding sites are listed in Supplementary Table 4.

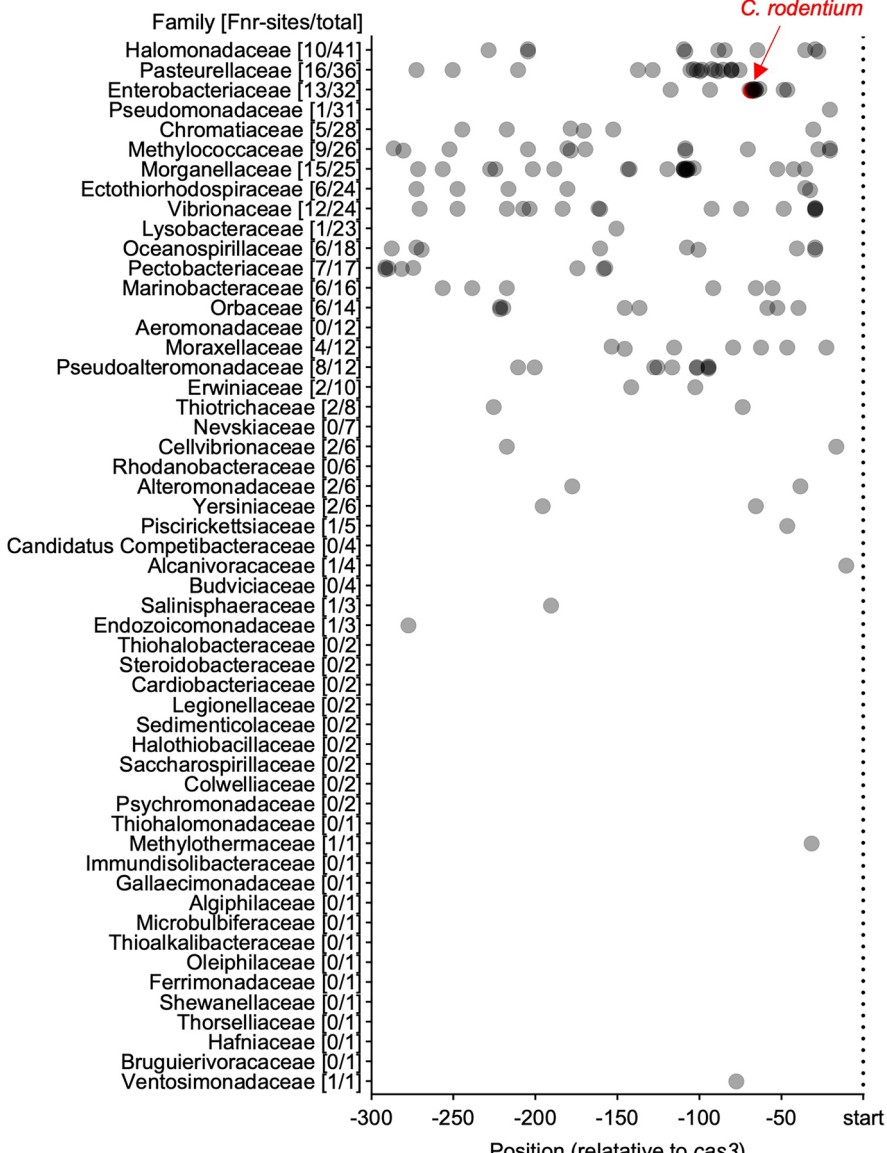

**Extended Data Fig. 4 | There is positional conservation of a putative Fnr binding site upstream of *cas3* in 13 of 32 Enterobacteriaceae.** The position of putative Fnr binding sites relative to 553 Gammaproteobacteria *cas3* orthologs, grouped by family. The number of genomes with at least one putative Fnr binding site in the 300 bp upstream of *cas3* (numerator) and the number of genomes analyzed within the family (denominator) are next to the family's name. Strains are listed in Supplementary Table 3 and putative Fnr binding sites are listed in Supplementary Table 4.

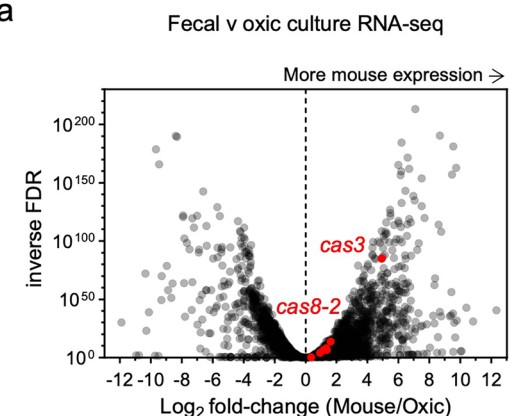

**a** Fecal v oxic culture RNA-seq

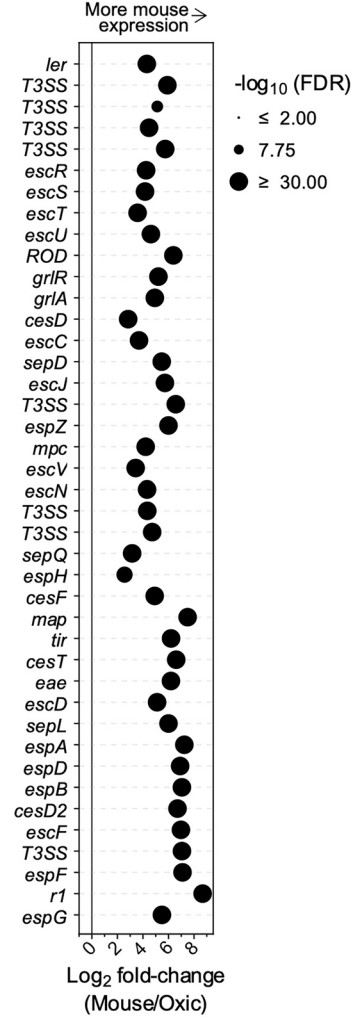

**b** Fecal v oxic culture RNA-seq
Locus of Enterocyte Effacement (LEE)

**Extended Data Fig. 5 | Extended presentation of RNA-seq data from Fig. 2.**
**a-b**, RNA-sequencing of *C. rodentium* recovered from the feces of infected, female, C57BL/6 J mice 7 days post inoculation compared to bacteria from oxic culture.

3 animal and 3 oxic samples. False discovery rate, FDR. **b**, The *C. rodentium* LEE virulence island was previously demonstrated to be transcribed in mice, but not in LB culture[29]. Additional data in Supplementary Table 1.

Matthew K. Waldor

# Reporting Summary

## Statistics

For all statistical analyses, confirm that the following items are present in the figure legend, table legend, main text, or Methods section.

| n/a | Confirmed | |
|---|---|---|
| ☐ | ☒ | The exact sample size (*n*) for each experimental group/condition, given as a discrete number and unit of measurement |
| ☐ | ☒ | A statement on whether measurements were taken from distinct samples or whether the same sample was measured repeatedly |
| ☐ | ☒ | The statistical test(s) used AND whether they are one- or two-sided<br>*Only common tests should be described solely by name; describe more complex techniques in the Methods section.* |
| ☒ | ☐ | A description of all covariates tested |
| ☐ | ☒ | A description of any assumptions or corrections, such as tests of normality and adjustment for multiple comparisons |
| ☐ | ☒ | A full description of the statistical parameters including central tendency (e.g. means) or other basic estimates (e.g. regression coefficient) AND variation (e.g. standard deviation) or associated estimates of uncertainty (e.g. confidence intervals) |
| ☐ | ☒ | For null hypothesis testing, the test statistic (e.g. *F*, *t*, *r*) with confidence intervals, effect sizes, degrees of freedom and *P* value noted<br>*Give P values as exact values whenever suitable.* |
| ☒ | ☐ | For Bayesian analysis, information on the choice of priors and Markov chain Monte Carlo settings |
| ☒ | ☐ | For hierarchical and complex designs, identification of the appropriate level for tests and full reporting of outcomes |
| ☒ | ☐ | Estimates of effect sizes (e.g. Cohen's *d*, Pearson's *r*), indicating how they were calculated |

*Our web collection on statistics for biologists contains articles on many of the points above.*

## Software and code

Policy information about availability of computer code

| | |
|---|---|
| Data collection | Transposon insertion sequencing data were collected using an Illumina NextSeq 1000. RNA-sequencing data were collected with an Illumina NovaSeq X Plus. Sequencing data demultiplexing, quality control, and adapter trimmer were performed with Illumina bcl convert. Quantitative PCR was performed using an Applied Biosystems StepOnePlus Real-Time PCR System. |
| Data analysis | Data analysis was performed using CLC genomics workbench (version 24.0.1), GraphPad Prism (version 10.4.1), Python (version 3.12), and Microsoft Excel. Custom scripts for Tn-seq analysis were originally developed in Basta & Campbell et. al., 2025 and are deposited online at https://github.com/dbasta27/InducTn-seq. |

For manuscripts utilizing custom algorithms or software that are central to the research but not yet described in published literature, software must be made available to editors and reviewers. We strongly encourage code deposition in a community repository (e.g. GitHub). See the Nature Portfolio guidelines for submitting code & software for further information.

## Data

Policy information about availability of data

All manuscripts must include a data availability statement. This statement should provide the following information, where applicable:
- Accession codes, unique identifiers, or web links for publicly available datasets
- A description of any restrictions on data availability
- For clinical datasets or third party data, please ensure that the statement adheres to our policy

RNA-seq and InducTn-seq sequencing reads are deposited in the Sequencing Read Archive (SRA) under accession no. PRJNA1254768. Reads were mapped to the C. rodentium ICC168 genome FN543502.1. Source data are provided with this paper as a source data file.

## Research involving human participants, their data, or biological material

Policy information about studies with human participants or human data. See also policy information about sex, gender (identity/presentation), and sexual orientation and race, ethnicity and racism.

| | |
|---|---|
| Reporting on sex and gender | N/A |
| Reporting on race, ethnicity, or other socially relevant groupings | N/A |
| Population characteristics | N/A |
| Recruitment | N/A |
| Ethics oversight | N/A |

Note that full information on the approval of the study protocol must also be provided in the manuscript.

# Field-specific reporting

Please select the one below that is the best fit for your research. If you are not sure, read the appropriate sections before making your selection.

☒ Life sciences          ☐ Behavioural & social sciences          ☐ Ecological, evolutionary & environmental sciences

For a reference copy of the document with all sections, see nature.com/documents/nr-reporting-summary-flat.pdf

# Life sciences study design

All studies must disclose on these points even when the disclosure is negative.

| | |
|---|---|
| Sample size | Sample sizes were determined by the variation observed in previous studies and preliminary experiments. |
| Data exclusions | No data were excluded from this study. |
| Replication | All attempts at replication were successful. Replicates are described in figure legends. |
| Randomization | Cages of animals were randomly assigned to experimental groups. Animals were not randomized within a cage due to the possibility of transmission between mice. |
| Blinding | Blinding was not used in these studies because the quantifiable variables (RNA-sequencing and plasmid exclusion) are not subject to investigator bias. |

# Reporting for specific materials, systems and methods

We require information from authors about some types of materials, experimental systems and methods used in many studies. Here, indicate whether each material, system or method listed is relevant to your study. If you are not sure if a list item applies to your research, read the appropriate section before selecting a response.

## Materials & experimental systems

| n/a | Involved in the study |
|-----|----------------------|
| ☒ | Antibodies |
| ☒ | Eukaryotic cell lines |
| ☒ | Palaeontology and archaeology |
| ☒ | Animals and other organisms |
| ☒ | Clinical data |
| ☒ | Dual use research of concern |
| ☒ | Plants |

## Methods

| n/a | Involved in the study |
|-----|----------------------|
| ☒ | ChIP-seq |
| ☒ | Flow cytometry |
| ☒ | MRI-based neuroimaging |

# Animals and other research organisms

Policy information about studies involving animals; ARRIVE guidelines recommended for reporting animal research, and Sex and Gender in Research

| | |
|---|---|
| Laboratory animals | Adult (9–12 weeks) C57BL/6J mice were purchased from Jackson Laboratory (strain #000664). |
| Wild animals | No wild animals were used in this study. |
| Reporting on sex | RNA-sequencing was performed on a single group of 3 female mice. RNA-sequencing data were confirmed and expanded upon with a plasmid exclusion assay, which was performed using an equal mix of male and female mice. No difference was observed between sexes and disaggregated data will be included in source data. |
| Field-collected samples | No field-collected samples were used in this study. |
| Ethics oversight | Animal studies were conducted at Brigham and Women's Hospital in compliance with the 'Guide for the Care and Use of Laboratory Animals' and according to protocols reviewed and approved by the Brigham and Women's Hospital's Institutional Animal Care and Use Committee under protocol 2016N000416. |

Note that full information on the approval of the study protocol must also be provided in the manuscript.

# Plants

| | |
|---|---|
| Seed stocks | N/A |
| Novel plant genotypes | N/A |
| Authentication | N/A |

