## [Peer Review File · Nature Microbiology]

Anoxia activates CRISPR-Cas immunity in the mouse intestine

Corresponding Author: Dr Ian Campbell

Version 0:

Reviewer comments:

Reviewer #1

(Remarks to the Author)

In this brief communication, Campbell and Basta and co-authors examine oxygen as a regulator of CRISPR-Cas gene activation and activity in *C. rodentium*. They find that the Cas locus is transcriptionally activated in anoxic conditions, and this is confirmed with functional assays of its activity. They further use a Tn-seq mutant library screen to identify the anaerobic transcription factor Fnr as required for Cas activation and function, and confirm this in vivo as well. Fnr regulation is also predicted in many other Enterobacteriaceae and other Proteobacteria. Thus, they propose that anaerobic activation of CRISPR defenses might be a common feature of bacteria that are likely to enter anaerobic environments such as the gut where attack by phages will be likely. To my knowledge this is an interesting and novel finding and a useful addition to this field. I don't see any major flaws that would prevent publication in Nature Microbiology.

Some minor points:

Fig. 1c, f, g there are no statistics shown, and I appreciate that. These are perfectly clear differences; statistical testing would not help anyone interpret the result.

Line 36: Plasmid acceptance or retention assays have been used for a long time to measure CRISPR activity. The authors should cite one of the earlier instances and/or explain why their assay is novel.

Please define the points (mean or median) and error (SD or SEM) shown in Fig. 2b.

Reviewer #2

(Remarks to the Author)

In this short research manuscript, Campbell and colleagues elegantly demonstrate that anoxic conditions lead to the expression and activity of CRISPR-Cas immunity of *Citrobacter rodentium*, an Enterobacteriaceae common to mammalian guts. The premise is simple and the rationale clear. The conclusions are supported by all the data presented, which include a combination of sequencing, in silico characterization, oxic and anoxic culturing, plasmid retention assays, and the use of transposon mutants. This was a very enjoyable and straightforward read and I only have minor comments to make.

Minor comments:

- L26-29: references to support this important argument should be added.
- Why is panel 1.F before panels 1.D and E in the main text? The panels should probably be re-ordered.
- L103-104: I'm not sure I follow why anoxia correlates with increased levels of phages and other mobile genetic elements. In addition to the local anoxia caused by increased microbial metabolism, anoxia would also suggest to the bacteria that they are in mucosal layers. These mucosal layers have been shown to be rich in mucins, to which phages have been shown to adhere to in vitro. This is speculative, but I understand this more than why there should be more phages in anoxic conditions in general.
- Fig 1H: it could be interesting to highlight the bacterial taxa that are common to mammalian guts. But this is not a required change.

Reviewer #3

(Remarks to the Author)

Summary

C. rodentium is a facultative anaerobe. RNA sequencing of *C. rodentium* in oxic and anoxic conditions revealed differential expression of genes involved in CRISPR-Cas immunity. Campbell & Basta et al. show that the native Type I-E CRISPR-Cas system of *C. rodentium* is only expressed (RNA-seq) and active (plasmid-targeting activity) under anoxic conditions. Subsequent Tn-Seq analysis suggests that the cas genes as well as fnr, a gene encoding an oxygen-responsive transcriptional regulator, are essential for effective CRISPR-Cas immunity under anoxic conditions. Individual knockout mutants verify the involvement of Fnr and the putative Fnr binding site in CRISPR-Cas immunity under anoxic conditions. Conservation of the Fnr binding site would

suggest that this type of cas3 regulation is widespread among Enterobacteriaceae. The authors hypothesize that this type of regulation is to ensure active immunity in the microbially rich anoxic mammalian intestine and show that plasmid retention in vivo is indeed increased when fnr is removed.

General comments

Regulation of CRISPR-Cas immunity in native settings is indeed a poorly understood topic. This brief communication provides a nice new insight, and I would recommend publishing the story after some minor revisions.

Major point

The use of outdated nomenclature, the absence of any kind of description of the mechanism of Type I-E CRISPR-Cas systems, and the general lack of references to ongoing work in the field suggest little effort on exploring existing literature. I understand that this is a very short format article, but I would like to see an awareness and better understanding of the biology of Type I-E CRISPR-Cas systems. There are tons of labs that do great work on these systems, examples would be the teams of Dr. Blake Wiedenheft, Dr. Stan Brouns, Dr. Luciano Marraffini, Dr. Dipali Sashital, and many others.

Minor points

- I suggest adding a schematic of the genomic region of the CRISPR-Cas system. First to familiarize readers with the genes of Type I-E systems, and second, to showcase the genomic architecture of the system (I'm guessing that like for many type I-E systems the Cascade genes are positioned close together or even overlapping while there is more spacing for regulatory elements around cas1/2 and cas3).
- Figure 1a. The nomenclature of the cas genes (CasA-E) is outdated. Although not incorrect, the scientific community moved to a unified naming scheme in ~2011. It would be more appropriate to use the current nomenclature (Cas6, Cas7, etc.).
- Figure 1a. Having a reference gene included in the panel (known from the literature to be regulated independent of oxic/anoxic conditions or known to be expressed under oxic conditions only) would showcase the technical rigor of this experiment.
- Table S1. Other defense systems also seem to be enriched, worth mentioning?
- Figure 1b. This doesn't actually provide any information on the plasmid retention assay. Wouldn't hurt to add a schematic overview of the plasmid retention assay (in the supplement).
- Methods. It would be nice to have the actual spacer/protospacer sequences to assess whether the control plasmid is appropriate. Only a single protospacer was tested? Which spacer from the native CRISPR array was selected and why?
- Table S2. Very nice to see that all Cascade genes + cas3 are highly enriched, and not Cas1/2, might be worth highlighting.
- Line 56. If Fnr is so widely conserved there should be information on other genes that are regulated by it. Add one of these as a positive control in your figures.
- Line 62. Please add context (references) for the prediction of the Fnr binding site, as well as for the specific mutation that was generated in this sequence.
- Methods line 133. There is no information on how the tree was actually constructed (iTOL is only used for visualization purposes).
- Figure S1/S2. It would be good to include a consensus logo of the predicted Fnr binding sites somewhere with supplementary figures 1 and 2. Could even add a consensus logo of the conserved Fnr binding sites that are not CRISPR-Cas related to strengthen the claim.
- Line 82. Do you believe that both Cas3 and Cascade are regulated by Fnr? This would be a good time to discuss the differences between cas3 and the other cas genes (could refer to the schematic of the CRISPR-Cas genes that was suggested before, would add the location of the Fnr binding site in there too).
- Figure 2a. Same comments as with figure 1a: nomenclature, and reference gene(s).
- Line 99. Good opportunity to mention the role of nucleoid-associated proteins studied in regulation of CRISPR-Cas systems.
- Line 101. If you're going to discuss the microbially rich environment that necessitates the activation of anti-MGE defense, then this might be a good time to mention other work in the field of CRISPR-Cas regulation by quorum-sensing.

Decision Letter:

16th July 2025

Dear Ian,

Thank you for your patience while your manuscript "Anoxia activates CRISPR-Cas immunity in the intestine" was under peer-review at Nature Microbiology. I'm very sorry this took quite so long to secure referees...but despite the delay, I'm thrilled to be able to return some good news. Your paper has now been seen by 3 referees, whose expertise and comments you will find at the of this email. You will see from their very positive comments below that they find your work of considerable interest. They do, however, raise some important points, so while we are very interested in the possibility of publishing your study in Nature Microbiology, we would like to consider your response to these concerns in the form of a revised manuscript before we make a final decision on publication.

Overall, the referees' reports are very clear and the remaining issues should be straightforward to address.

If you have not done so already please begin to revise your manuscript so that it conforms to our Brief Communication format instructions at <http://www.nature.com/nmicrobiol/info/final-submission/>

Our normal length limit for a Brief Communication to Nature Microbiology with no more than 2 small display items (figures or tables) is 1,250 words. We have some flexibility, and can allow a revised manuscript at 1,500 words, but please consider this a firm upper limit.

Some reduction could be achieved by focusing any introductory material and moving it to the start of your opening 'bold' paragraph, whose function is to outline the background to your work, describe in a sentence your new observations, and explain your main conclusions. The discussion should also be limited. Methods should be described in a separate section following the discussion, we do not place a word limit on Methods.

Nature Microbiology titles should give a sense of the main new findings of a manuscript, and should not contain punctuation. Please keep in mind that we strongly discourage active verbs in titles, and that they should ideally fit within 90 characters each (including spaces).

Please include a data availability statement as a separate section after Methods but before references, under the heading "Data Availability". This section should inform readers about the availability of the data used to support the conclusions of your study. This information includes accession codes to public repositories (data banks for protein, DNA or RNA sequences, microarray, proteomics data etc...), references to source data published alongside the paper, unique identifiers such as URLs to data repository entries, or data set DOIs, and any other statement about data availability. At a minimum, you should include the following statement: "The data that support the findings of this study are available from the corresponding author upon request", mentioning any restrictions on availability. If DOIs are provided, we also strongly encourage including these in the Reference list (authors, title, publisher (repository name), identifier, year). For more guidance on how to write this section please see: <http://www.nature.com/authors/policies/data/data-availability-statements-data-citations.pdf>

To improve the accessibility of your paper to readers from other research areas, please pay particular attention to the wording of the paper's opening bold paragraph, which serves both as an introduction and as a brief, non-technical summary in about 150 words. If, however, you require one or two extra sentences to explain your work clearly, please include them even if the paragraph is over-length as a result. The opening paragraph should not contain references. Because scientists from other sub-disciplines will be interested in your results and their implications, it is important to explain essential but specialised terms concisely. We suggest you show your summary paragraph to colleagues in other fields to uncover any problematic concepts.

If your paper is accepted for publication, we will edit your display items electronically so they conform to our house style and will reproduce clearly in print. If necessary, we will re-size figures to fit single or double column width. If your figures contain several parts, the parts should form a neat rectangle when assembled. Choosing the right electronic format at this stage will speed up the processing of your paper and give the best possible results in print. We would like the figures to be supplied as vector files - EPS, PDF, AI or postscript (PS) file formats (not raster or bitmap files), preferably generated with vector-graphics software (Adobe Illustrator for example). Please try to ensure that all figures are non-flattened and fully editable. All images should be at least 300 dpi resolution (when figures are scaled to approximately the size that they are to be printed at) and in RGB colour format. Please do not submit Jpeg or flattened TIFF files. Please see also 'Guidelines for Electronic Submission of Figures' at the end of this letter for further detail.

Figure legends must provide a brief description of the figure and the symbols used, within 350 words, including definitions of any error bars employed in the figures.

Please include a statement before the acknowledgements naming the author to whom correspondence and requests for materials should be addressed.

Finally, we require authors to include a statement of their individual contributions to the paper -- such as experimental work, project planning, data analysis, etc. -- immediately after the acknowledgements. The statement should be short, and refer to

authors by their initials. For details please see the Authorship section of our joint Editorial policies at http://www.nature.com/authors/editorial_policies/authorship.html

* include a point-by-point response to any editorial suggestions and to our referees. Please include your response to the editorial suggestions in your cover letter, and please upload your response to the referees as a separate document.

* ensure it complies with our format requirements for Letters as set out in our guide to authors at www.nature.com/nmicrobiol/info/gta/

* state in a cover note the length of the text, methods and legends; the number of references; number and estimated final size of figures and tables

* resubmit electronically if possible using the link below to access your home page:

Link Redacted

*This url links to your confidential homepage and associated information about manuscripts you may have submitted or be reviewing for us. If you wish to forward this e-mail to co-authors, please delete this link to your homepage first.

Please ensure that all correspondence is marked with your Nature Microbiology reference number in the subject line.

Nature Microbiology is committed to improving transparency in authorship. As part of our efforts in this direction, we are now requesting that all authors identified as 'corresponding author' on published papers create and link their Open Researcher and Contributor Identifier (ORCID) with their account on the Manuscript Tracking System (MTS), prior to acceptance. This applies to primary research papers only. ORCID helps the scientific community achieve unambiguous attribution of all scholarly contributions. You can create and link your ORCID from the home page of the MTS by clicking on 'Modify my Springer Nature account'. For more information please visit www.springernature.com/orcid.

We hope to receive your revised paper within three weeks. If you cannot send it within this time, please let us know.

Yours sincerely,

Reviewer Expertise:

Referee #1: host-microbiome; Citrobacter; mouse models

Referee #2: gut microbiome; microbial ecology

Referee #3: CRISPR

Reviewers Comments:

Reviewer #1 (Remarks to the Author):

In this brief communication, Campbell and Basta and co-authors examine oxygen as a regulator of CRISPR-Cas gene activation and activity in *C. rodentium*. They find that the Cas locus is transcriptionally activated in anoxic conditions, and this is confirmed with functional assays of its activity. They further use a Tn-seq mutant library screen to identify the anaerobic transcription factor Fnr as required for Cas activation and function, and confirm this in vivo as well. Fnr regulation is also predicted in many other Enterobacteriaceae and other Proteobacteria. Thus, they propose that anaerobic activation of CRISPR defenses might be a common feature of bacteria that are likely to enter anaerobic environments such as the gut where attack by phages will be likely. To my knowledge this is an interesting and novel finding and a useful addition to this field. I don't see any major flaws that would prevent publication in Nature Microbiology.

Some minor points:

Fig. 1c, f, g there are no statistics shown, and I appreciate that. These are perfectly clear differences; statistical testing would not help anyone interpret the result.

Line 36: Plasmid acceptance or retention assays have been used for a long time to measure CRISPR activity. The authors should cite one of the earlier instances and/or explain why their assay is novel.

Please define the points (mean or median) and error (SD or SEM) shown in Fig. 2b.

Reviewer #2 (Remarks to the Author):

In this short research manuscript, Campbell and colleagues elegantly demonstrate that anoxic conditions lead to the expression and activity of CRISPR-Cas immunity of *Citrobacter rodentium*, an Enterobacteriaceae common to mammalian guts. The premise is simple and the rationale clear. The conclusions are supported by all the data presented, which include a combination of sequencing, in silico characterization, oxic and anoxic culturing, plasmid retention assays, and the use of transposon mutants. This was a very enjoyable and straightforward read and I only have minor comments to make.

Minor comments:

- L26-29: references to support this important argument should be added.
- Why is panel 1.F before panels 1.D and E in the main text? The panels should probably be re-ordered.
- L103-104: I'm not sure I follow why anoxia correlates with increased levels of phages and other mobile genetic elements. In addition to the local anoxia caused by increased microbial metabolism, anoxia would also suggest to the bacteria that they are in mucosal layers. These mucosal layers have been shown to be rich in mucins, to which phages have been shown to adhere to in vitro. This is speculative, but I understand this more than why there should be more phages in anoxic conditions in general.
- Fig 1H: it could be interesting to highlight the bacterial taxa that are common to mammalian guts. But this is not a required change.

Reviewer #3 (Remarks to the Author):

Summary

C. rodentium is a facultative anaerobe. RNA sequencing of *C. rodentium* in oxic and anoxic conditions revealed differential expression of genes involved in CRISPR-Cas immunity. Campbell & Basta et al. show that the native Type I-E CRISPR-Cas system of *C. rodentium* is only expressed (RNA-seq) and active (plasmid-targeting activity) under anoxic conditions. Subsequent Tn-Seq analysis suggests that the *cas* genes as well as *fnr*, a gene encoding an oxygen-responsive transcriptional regulator, are essential for effective CRISPR-Cas immunity under anoxic conditions. Individual knockout mutants verify the involvement of *Fnr* and the putative *Fnr* binding site in CRISPR-Cas immunity under anoxic conditions. Conservation of the *Fnr* binding site would suggest that this type of *cas3* regulation is widespread among Enterobacteriaceae. The authors hypothesize that this type of regulation is to ensure active immunity in the microbially rich anoxic mammalian intestine and show that plasmid retention in vivo is indeed increased when *fnr* is removed.

General comments

Regulation of CRISPR-Cas immunity in native settings is indeed a poorly understood topic. This brief communication provides a nice new insight, and I would recommend publishing the story after some minor revisions.

Major point

The use of outdated nomenclature, the absence of any kind of description of the mechanism of Type I-E CRISPR-Cas systems, and the general lack of references to ongoing work in the field suggest little effort on exploring existing literature. I understand that this is a very short format article, but I would like to see an awareness and better understanding of the biology of Type I-E CRISPR-Cas systems. There are tons of labs that do great work on these systems, examples would be the teams of Dr. Blake Wiedenheft, Dr. Stan Brouns, Dr. Luciano Marraffini, Dr. Dipali Sashital, and many others.

Minor points

- I suggest adding a schematic of the genomic region of the CRISPR-Cas system. First to familiarize readers with the genes of Type I-E systems, and second, to showcase the genomic architecture of the system (I'm guessing that like for many type I-E systems the Cascade genes are positioned close together or even overlapping while there is more spacing for regulatory elements around *cas1/2* and *cas3*).
- Figure 1a. The nomenclature of the *cas* genes (*CasA-E*) is outdated. Although not incorrect, the scientific community moved to a unified naming scheme in ~2011. It would be more appropriate to use the current nomenclature (*Cas6*, *Cas7*, etc.).
- Figure 1a. Having a reference gene included in the panel (known from the literature to be regulated independent of oxic/anoxic conditions or known to be expressed under oxic conditions only) would showcase the technical rigor of this experiment.
- Table S1. Other defense systems also seem to be enriched, worth mentioning?
- Figure 1b. This doesn't actually provide any information on the plasmid retention assay. Wouldn't hurt to add a schematic overview of the plasmid retention assay (in the supplement).
- Methods. It would be nice to have the actual spacer/protospacer sequences to assess whether the control plasmid is appropriate. Only a single protospacer was tested? Which spacer from the native CRISPR array was selected and why?
- Table S2. Very nice to see that all Cascade genes + *cas3* are highly enriched, and not *Cas1/2*, might be worth highlighting.
- Line 56. If *Fnr* is so widely conserved there should be information on other genes that are regulated by it. Add one of these as a positive control in your figures.
- Line 62. Please add context (references) for the prediction of the *Fnr* binding site, as well as for the specific mutation that was generated in this sequence.

- Methods line 133. There is no information on how the tree was actually constructed (iTOL is only used for visualization purposes).

- Figure S1/S2. It would be good to include a consensus logo of the predicted Fnr binding sites somewhere with supplementary figures 1 and 2. Could even add a consensus logo of the conserved Fnr binding sites that are not CRISPR-Cas related to strengthen the claim.

- Line 82. Do you believe that both Cas3 and Cascade are regulated by Fnr? This would be a good time to discuss the differences between cas3 and the other cas genes (could refer to the schematic of the CRISPR-Cas genes that was suggested before, would add the location of the Fnr binding site in there too).

- Figure 2a. Same comments as with figure 1a: nomenclature, and reference gene(s).

- Line 99. Good opportunity to mention the role of nucleoid-associated proteins studied in regulation of CRISPR-Cas systems.

- Line 101. If you're going to discuss the microbially rich environment that necessitates the activation of anti-MGE defense, then this might be a good time to mention other work in the field of CRISPR-Cas regulation by quorum-sensing.

Version 1:

Reviewer comments:

Reviewer #3

(Remarks to the Author)

Thanks for incorporating the feedback. I have no further comments and wish the authors the best with the remainder of the publication process.

Decision Letter:

Our ref: NMICROBIOL-25041528A

9th September 2025

Dear Ian,

Once again, we appreciate your patience here... Thank you for submitting your revised manuscript "Anoxia activates CRISPR-Cas immunity in the intestine" (NMICROBIOL-25041528A). It has now been seen by the original referees and their comments are below. The reviewers find that the paper has improved in revision, and therefore we'll be happy in principle to publish it in Nature Microbiology, pending minor revisions to satisfy the referees' final requests and to comply with our editorial and formatting guidelines.

Thank you again for your interest in Nature Microbiology Please do not hesitate to contact me if you have any questions.

Sincerely,

Reviewer #3 (Remarks to the Author):

Thanks for incorporating the feedback. I have no further comments and wish the authors the best with the remainder of the publication process.

Version 2:

Decision Letter:

2nd October 2025

Dear Dr Campbell,

I am pleased to accept your Brief Communication "Anoxia activates CRISPR-Cas immunity in the mouse intestine" for publication in Nature Microbiology. Thank you for having chosen to submit your work to us and many congratulations.

Authors may need to take specific actions to achieve compliance with funder and institutional open access mandates. If your research is supported by a funder that requires immediate open access (e.g. according to [Plan S principles](https://www.springernature.com/gp/open-science/plan-s-compliance) or the [NIH public access policy](https://www.springernature.com/gp/open-science/us-federal-agency-compliance)) then you should select the gold OA route, and we will direct you to the compliant route where possible. Because authors warrant under our subscription licensing terms that they haven't committed to licensing any version of their article under a licence inconsistent with the terms of our agreement – including the applicable embargo period – publication under the subscription model isn't suitable for authors whose funders require no embargo.

With kind regards,

P.S. Click on the following link if you would like to recommend Nature Microbiology to your librarian
<http://www.nature.com/subscriptions/recommend.html#forms>

** Visit the Springer Nature Editorial and Publishing website at http://editorial-jobs.springernature.com?utm_source=ejP_NMicro_email&utm_medium=ejP_NMicro_email&utm_campaign=ejp_NMicro for more information about our career opportunities. If you have any questions please click [here](mailto:editorial.publishing.jobs@springernature.com). **

Response to Referees

As outlined below in our point-by-point responses, we have addressed all the referees' comments with new analyses and modifications to the text. These modifications include the addition of new references, an extended discussion of the CRISPR-Cas literature, a new figure panel, and three new supplementary figures. The referees' thoughtful questions and suggestions have improved the manuscript, and we thank you for your time and efforts in considering our work. We look forward to the assessment of our revised manuscript.

Point-by-point response (review in black, response in blue)

Reviewer #1 (Remarks to the Author):

In this brief communication, Campbell and Basta and co-authors examine oxygen as a regulator of CRISPR-Cas gene activation and activity in *C. rodentium*. They find that the Cas locus is transcriptionally activated in anoxic conditions, and this is confirmed with functional assays of its activity. They further use a Tn-seq mutant library screen to identify the anaerobic transcription factor Fnr as required for Cas activation and function, and confirm this in vivo as well. Fnr regulation is also predicted in many other Enterobacteriaceae and other Proteobacteria. Thus, they propose that anaerobic activation of CRISPR defenses might be a common feature of bacteria that are likely to enter anaerobic environments such as the gut where attack by phages will be likely. To my knowledge this is an interesting and novel finding and a useful addition to this field. I don't see any major flaws that would prevent publication in Nature Microbiology.

We thank Reviewer 1 for their positive comments.

Some minor points:

Fig. 1c, f, g there are no statistics shown, and I appreciate that. These are perfectly clear differences; statistical testing would not help anyone interpret the result.

We strongly agree.

Line 36: Plasmid acceptance or retention assays have been used for a long time to measure CRISPR activity. The authors should cite one of the earlier instances and/or explain why their assay is novel.

Thank you, we have added a reference to Marraffini & Sontheimer 2008 when first describing our plasmid retention assay. To our knowledge, plasmid-based assays that measure CRISPR-Cas activity, including Marraffini & Sontheimer 2008, have relied on constitutive CRISPR-Cas expression that prevents initial plasmid uptake (see also PMID: 21813460, PMID: 22114197, PMID: 24039596, PMID: 26007654). Our assay is distinctive in that we monitor conditional CRISPR-Cas activity by measuring plasmid elimination rather than uptake.

Marraffini, L. A. & Sontheimer, E. J. CRISPR interference limits horizontal gene transfer in staphylococci by targeting DNA. *Science* 322, 1843–1845 (2008).

Please define the points (mean or median) and error (SD or SEM) shown in Fig. 2b.

Thank you for identifying this omission. The lines and shading indicate geometric mean and geometric SD, which we have now defined in the figure legend.

Reviewer #2 (Remarks to the Author):

In this short research manuscript, Campbell and colleagues elegantly demonstrate that anoxic conditions lead to the expression and activity of CRISPR-Cas immunity of *Citrobacter rodentium*, an Enterobacteriaceae common to mammalian guts. The premise is simple and the rationale clear. The conclusions are supported by all the data presented, which include a combination of sequencing, in silico characterization, oxic and anoxic culturing, plasmid retention assays, and the use of transposon mutants. This was a very enjoyable and straightforward read and I only have minor comments to make.

We thank Reviewer 2 for their positive comments.

Minor comments:

- L26-29: references to support this important argument should be added.

Thank you for identifying this omission. We have added the following references:

Brouns, S. J. J. et al. Small CRISPR RNAs guide antiviral defense in prokaryotes. *Science* 321, 960–964 (2008).

Pougach, K. et al. Transcription, processing and function of CRISPR cassettes in *Escherichia coli*. *Mol Microbiol* 77, 1367–1379 (2010).

Westra, E. R. et al. H-NS-mediated repression of CRISPR-based immunity in *Escherichia coli* K12 can be relieved by the transcription activator LeuO. *Mol Microbiol* 77, 1380–1393 (2010).

- Why is panel 1.F before panels 1.D and E in the main text? The panels should probably be re-ordered.

We have modified the order of the panels in Figure 1.

- L103-104: I'm not sure I follow why anoxia correlates with increased levels of phages and other mobile genetic elements. In addition to the local anoxia caused by increased microbial metabolism, anoxia would also suggest to the bacteria that they are in mucosal layers. These mucosal layers have been shown to be rich in mucins, to which phages have been shown to adhere to in vitro. This is speculative, but I understand this more than why there should be more phages in anoxic conditions in general.

Thank you for identifying this area for clarification. The original statement was overly speculative, and we have modified the text to simply highlight the correlation between the low oxygen environment of the intestine and its dense microbial community, including

phage. Our hypothesis is that a lack of oxygen signals that *C. rodentium* is in the intestine, where it is advantageous to activate CRISPR-Cas expression.

The hypothesis regarding mucin and phage is very interesting. However, from our understanding of intestinal physiology, there is an oxygen gradient that radiates from the epithelium, and thus the mucosal layers have relatively higher concentrations of oxygen compared to more luminal environments. A spatial study of CRISPR-Cas activity within the intestine is something we hope to carry out in the future.

- Fig 1H: it could be interesting to highlight the bacterial taxa that are common to mammalian guts. But this is not a required change.

Thank you for this interesting idea. Many of the Enterobacteriaceae that contain the Fnr-binding site have been observed in the mammalian intestine (e.g., *Escherichia*, *Citrobacter*, *Klebsiella*). We have modified the text on lines 77-78 to highlight this point.

Reviewer #3 (Remarks to the Author):

Summary

C. rodentium is a facultative anaerobe. RNA sequencing of *C. rodentium* in oxic and anoxic conditions revealed differential expression of genes involved in CRISPR-Cas immunity. Campbell & Basta et al. show that the native Type I-E CRISPR-Cas system of *C. rodentium* is only expressed (RNA-seq) and active (plasmid-targeting activity) under anoxic conditions. Subsequent Tn-Seq analysis suggests that the *cas* genes as well as *fnr*, a gene encoding an oxygen-responsive transcriptional regulator, are essential for effective CRISPR-Cas immunity under anoxic conditions. Individual knockout mutants verify the involvement of Fnr and the putative Fnr binding site in CRISPR-Cas immunity under anoxic conditions. Conservation of the Fnr binding site would suggest that this type of *cas3* regulation is widespread among Enterobacteriaceae. The authors hypothesize that this type of regulation is to ensure active immunity in the microbially rich anoxic mammalian intestine and show that plasmid retention in vivo is indeed increased when *fnr* is removed.

General comments

Regulation of CRISPR-Cas immunity in native settings is indeed a poorly understood topic. This brief communication provides a nice new insight, and I would recommend publishing the story after some minor revisions.

We thank Reviewer 3 for their positive comments.

Major point

The use of outdated nomenclature, the absence of any kind of description of the mechanism of Type I-E CRISPR-Cas systems, and the general lack of references to ongoing work in the field suggest little effort on exploring existing literature. I understand that this is a very short format article, but I would like to see an awareness and better understanding of the biology of Type I-E CRISPR-Cas systems. There are tons of labs that do great work on these systems, examples would be the teams of Dr. Blake Wiedenheft, Dr. Stan Brouns, Dr. Luciano Marraffini, Dr. Dipali Sashital, and many others.

We strongly agree that excellent work is being done within the field and that this manuscript builds on the legacy of many groups. We have expanded our references to include more citations of recent, primary literature rather than relying on reviews. In doing so, we have approximately doubled the number of citations. Additions include:

Original manuscripts discussing the lack of CRISPR-Cas activity in Enterobacteriaceae in standard culture conditions.

Brouns, S. J. J. et al. Small CRISPR RNAs guide antiviral defense in prokaryotes. *Science* 321, 960–964 (2008).

Pougach, K. et al. Transcription, processing and function of CRISPR cassettes in *Escherichia coli*. *Mol Microbiol* 77, 1367–1379 (2010).

Westra, E. R. et al. H-NS-mediated repression of CRISPR-based immunity in *Escherichia coli* K12 can be relieved by the transcription activator LeuO. *Mol Microbiol* 77, 1380–1393 (2010).

The early use of a CRISPR-Cas reporter plasmid.

Marraffini, L. A. & Sontheimer, E. J. CRISPR interference limits horizontal gene transfer in staphylococci by targeting DNA. *Science* 322, 1843–1845 (2008).

Manuscripts discussing transcriptional regulation of CRISPR-Cas systems by H-NS.

Westra, E. R. et al. H-NS-mediated repression of CRISPR-based immunity in *Escherichia coli* K12 can be relieved by the transcription activator LeuO. *Mol Microbiol* 77, 1380–1393 (2010).

Pul, Ü. et al. Identification and characterization of *E. coli* CRISPR-cas promoters and their silencing by H-NS. *Mol Microbiol* 75, 1495–1512 (2010).

Pougach, K. et al. Transcription, processing and function of CRISPR cassettes in *Escherichia coli*. *Mol Microbiol* 77, 1367–1379 (2010).

Recent manuscripts discussing the role of quorum sensing in regulating CRISPR-Cas activity.

Patterson, A. G. et al. Quorum Sensing Controls Adaptive Immunity through the Regulation of Multiple CRISPR-Cas Systems. *Mol Cell* 64, 1102–1108 (2016).

Høyland-Kroghsbo, N. M. et al. Quorum sensing controls the *Pseudomonas aeruginosa* CRISPR-Cas adaptive immune system. *Proc Natl Acad Sci U S A* 114, 131–135 (2017).

Maharajan, A. D., Hjerde, E., Hansen, H. & Willassen, N. P. Quorum Sensing Controls the CRISPR and Type VI Secretion Systems in *Aliivibrio wodanis* 06/09/139. *Front Vet Sci* 9, 799414 (2022).

Additionally, we have added a schematic of the *C. rodentium* CRISPR-Cas locus to Figure 1 and references to previous studies to better introduce the reader to the system.

Minor points

- I suggest adding a schematic of the genomic region of the CRISPR-Cas system. First to familiarize readers with the genes of Type I-E systems, and second, to showcase the genomic architecture of the system (I'm guessing that like for many type I-E systems the

Cascade genes are positioned close together or even overlapping while there is more spacing for regulatory elements around *cas1/2* and *cas3*).

Thank you for this suggestion. We have added a schematic of the *C. rodentium* CRISPR-Cas locus to the beginning of Figure 1. Like many Type I-E systems, the CRISPR-Cas locus of *C. rodentium* begins with *cas3* and contains a gap between *cas3* and the cascade genes. Cascade is followed immediately by *cas1/2*.

- Figure 1a. The nomenclature of the *cas* genes (CasA-E) is outdated. Although not incorrect, the scientific community moved to a unified naming scheme in ~2011. It would be more appropriate to use the current nomenclature (Cas6, Cas7, etc.).

Thank you for bringing the updated nomenclature to our attention. We have updated the nomenclature throughout the manuscript based on the classification proposed in Makarova et. al., 2020 (doi.org/10.1038/s41579-019-0299-x).

Original manuscript	Revised manuscript
cas3	cas3
casA	cas8
casB	cas11
casC	cas7
casD	cas5
casE	cas6
cas1	cas1
cas2	cas2

- Figure 1a. Having a reference gene included in the panel (known from the literature to be regulated independent of oxic/anoxic conditions or known to be expressed under oxic conditions only) would showcase the technical rigor of this experiment.

We have created a new supplementary figure that expands on the RNA-seq results (Supplementary figure 1). This figure shows the complete RNA-seq results and highlights genes previously characterized to be activated/repressed by Fnr under anoxic conditions, as well as the negative control used in our qPCR (*rpoA*).

- Table S1. Other defense systems also seem to be enriched, worth mentioning?

Thank you for the suggestion. We find that several *C. rodentium* phage defense systems have increased expression in anoxic vs oxic culture conditions in our RNA-seq dataset (e.g., RADAR, Lamassu, Gao Qat). However, the signal is weak (2-4 fold increase in anoxic conditions). The impact of these relatively minor changes in expression is unclear without further investigation. While these data may contain interesting avenues for future

investigation, we believe it is overly speculative to make a point of these data in the manuscript.

- Figure 1b. This doesn't actually provide any information on the plasmid retention assay. Wouldn't hurt to add a schematic overview of the plasmid retention assay (in the supplement).

We have replaced the schematic representation of the plasmid retention assay in Figure 1b with a more detailed schematic (new Supplementary Figure 2).

- Methods. It would be nice to have the actual spacer/protospacer sequences to assess whether the control plasmid is appropriate. Only a single protospacer was tested? Which spacer from the native CRISPR array was selected and why?

We have added the protospacer sequences to the methods and the new Supplementary Figure 2 assay schematic. *C. rodentium* has two CRISPR arrays, one upstream and the other downstream of the CRISPR-Cas locus. We tested all of the spacers in the upstream array and found that they all behaved similarly, except for a few spacers with repeats that do not match the consensus. We chose the spacer 'ATCTGTTTATAGCTGGCTATAAAATTATAAA', which had the strongest activity in this preliminary experiment for subsequent experiments.

- Table S2. Very nice to see that all Cascade genes + cas3 are highly enriched, and not Cas1/2, might be worth highlighting.

We likewise found these data to be highly specific. We have added a comment to the text of the manuscript indicating that the absence of *cas1* and *cas2* from the screen hits indicates specificity.

- Line 56. If Fnr is so widely conserved there should be information on other genes that are regulated by it. Add one of these as a positive control in your figures.

Thank you for the suggestion. In the RNA-seq data in the new Supplementary Figure 1 we have highlighted genes regulated by Fnr in response to anoxia in *E. coli* (<https://doi.org/10.1371/journal.pgen.1003565>).

- Line 62. Please add context (references) for the prediction of the Fnr binding site, as well as for the specific mutation that was generated in this sequence.

Thank you for identifying this omission. We have added a reference to the motif enrichment software that we used to find the Fnr-motif upstream of *cas3* and to the original work that defined the motif in *E. coli*. The Fnr-binding site mutation was chosen because the first 2 bp of the Fnr-motif are highly conserved in the *E. coli* consensus sequence. We have added the *E. coli* Fnr-binding site consensus sequence to Figure 1 to clarify.

- Methods line 133. There is no information on how the tree was actually constructed (iTOL is only used for visualization purposes).

Thank you for identifying this omission. We used MAFFT (strategy: FFT-NS-2; model: DNA200; v7.526) to align the sequences and FastTree (model: Jukes-Cantor with CAT rate heterogeneity; v2.1.11) to construct the phylogeny. We have added these details to the methods.

- Figure S1/S2. It would be good to include a consensus logo of the predicted Fnr binding sites somewhere with supplementary figures 1 and 2. Could even add a consensus logo of the conserved Fnr binding sites that are not CRISPR-Cas related to strengthen the claim.

Thank you for the suggestion. We have added a consensus logo of the putative Enterobacteriaceae Fnr-binding sites upstream of *cas3* to Figure 1.

- Line 82. Do you believe that both Cas3 and Cascade are regulated by Fnr? This would be a good time to discuss the differences between *cas3* and the other *cas* genes (could refer to the schematic of the CRISPR-Cas genes that was suggested before, would add the location of the Fnr binding site in there too).

We think *cas3* and the Cascade genes are regulated separately; there is a gap between *cas3* and the rest of the *cas* locus (see new Figure 1 schematic), and within mice *cas3* transcripts diverge from the rest of the *cas* locus (Figure 2a). Given the transcriptional response of the Cascade genes to anoxia, it is possible that they are also regulated by Fnr, but we have not directly tested this hypothesis. Ultimately, CRISPR-Cas activity (plasmid retention) correlates with *cas3* expression, and it is therefore unclear whether the Cascade genes are constitutively expressed or only expressed in anoxic conditions.

To clarify this point we have added a new schematic of the *C. rodentium cas* locus to Figure 1, with the putative Fnr-binding site labeled upstream of *cas3*.

- Figure 2a. Same comments as with figure 1a: nomenclature, and reference gene(s).

As discussed above, we have changed the *cas* nomenclature throughout the manuscript. We have created a new supplemental figure to expand on the RNA-seq results comparing oxic culture to bacteria recovered from the feces of infected mice (new Supplemental Figure 5). This figure shows the complete RNA-seq results and highlights genes previously characterized to be transcribed in the murine host, but not in culture (the virulence island, the Locus of Enterocyte Effacement).

- Line 99. Good opportunity to mention the role of nucleoid-associated proteins studied in regulation of CRISPR-Cas systems.

Thank you for the suggestion. We have added a discussion of previous reports that H-NS regulates the transcription of CRISPR-Cas systems and added the following references.

Westra, E. R. et al. H-NS-mediated repression of CRISPR-based immunity in *Escherichia coli* K12 can be relieved by the transcription activator LeuO. *Mol Microbiol* 77, 1380–1393 (2010).

Pul, Ü. et al. Identification and characterization of *E. coli* CRISPR-cas promoters and their silencing by H-NS. *Mol Microbiol* 75, 1495–1512 (2010).

Pougach, K. et al. Transcription, processing and function of CRISPR cassettes in *Escherichia coli*. *Mol Microbiol* 77, 1367–1379 (2010).

- Line 101. If you're going to discuss the microbially rich environment that necessitates the activation of anti-MGE defense, then this might be a good time to mention other work in the field of CRISPR-Cas regulation by quorum-sensing.

Thank you for the suggestion. We have added a discussion of previous reports that quorum sensing regulates CRISPR-Cas activity and added the following references.

Patterson, A. G. et al. Quorum Sensing Controls Adaptive Immunity through the Regulation of Multiple CRISPR-Cas Systems. *Mol Cell* 64, 1102–1108 (2016).

Høyland-Kroghsbo, N. M. et al. Quorum sensing controls the *Pseudomonas aeruginosa* CRISPR-Cas adaptive immune system. *Proc Natl Acad Sci U S A* 114, 131–135 (2017).

Maharajan, A. D., Hjerde, E., Hansen, H. & Willassen, N. P. Quorum Sensing Controls the CRISPR and Type VI Secretion Systems in *Aliivibrio wodanis* 06/09/139. *Front Vet Sci* 9, 799414 (2022).